

# Ionization efficiency at sub-keV energies for crystals and noble liquids

**Youssef Sarkis⋆, Alexis A. Aguilar-Arevalo and Juan Carlos D'Olivo**

Instituto de Ciencias Nucleares UNAM

⋆ youssef@ciencias.unam.mx

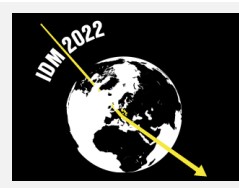

## Abstract

We study the ionization and light yields produced by nuclear recoils at low energies in pure crystals and noble liquids in the context of Lindhard's integral equation, incorporating the effects of binding energy, improved modeling of the electronic stopping, and electronic straggling. We consider three different models for the electronic stopping power that incorporate Coulomb repulsion effects at low energies, and Bohr electronic stripping for high energies. Finally, we discuss possible new effects near threshold.

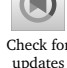

# 1 Introduction

Dark Matter, Neutrinos and other rare searches [1] required to be detected by direct sub-keV recoil energy spectrum from terrestrial and astrophysical sources. For pure element ionization detectors like Si [2], Ge [3], liquid Argon (LAr) [4] and liquid Xenon (LXe) [5], the typical recoil energy analyzed is less than 1 keV, where at this regime only a fraction of this energy goes to ionization. The convertion of the total recoil energy to visible or electronic energy is given by the ionization efficiency or quenching factor. Lindhard integral equation with binding energy [6] already have proven to success in describing low energy measurments for ionization efficiency in silicon [7], this rediscovered approach can be used to obtained the ionization deposit energy that an non ionizable particle gives to a pure media when interacts with a single nuclei, by separating electronic from nuclear processes.

Although Lindhard integral equation[1] with constant binding energy describes accurately the data of silicon below 4 keV, it required an outranged value from the expected one, Frenkel pair energy [8]. Furthermore, other studies [9, 10] evidenciate that the electronic stopping power computed by Lindhard [11] might be overestimated at low energies.

For DM or CE$\nu$NS searches with Si and Ge ionization detectors the ionization efficiency (quenching factor) plays an important role for calibration. Different quenching factor leads to different shape of energy spectrum specially near threshold (100 eV$_{ee}$). Quenching factor has the effect to move the events of the spectrum to low energy, where some of them can pass the threshold. This lead to a systematic error fluctuation in low energy detection experiments, where usually spectrum rate for different quenching factor models are reported, [12, 13].

# 2 Ionization and Lindhard Integral Equation

When a non ionizing particle, like DM or a neutrino, interacts with a crystal detector it deposit a total recoil energy $E_R$. When this happens part of the recoil energy is used to disrupt the atomic-binding $U$, so then the ion now moves with an kinetic energy $E$ (small than $E_R$). This process continue until the energy of the ion is not enough to disrupt the atomic binding. In this work we are going to use Lindhard units $\varepsilon = CE$, where $C = 11.5/Z^{7/3}(1/\text{keV})$. The energy is divided into atomic motion ($\bar{\nu}$) due to all subsequent ions collisions and ionization energy due to excite electrons ($\bar{\eta}$). So we define the ionization efficiency or quenching factor by the quotient of the energy that goes to ionization over the total recoil energy deposited in the material,

$$f_n = \frac{\bar{\eta}}{\varepsilon_R}. \tag{1}$$

We are going to assume as work hypothesis that the ionization occurs only when the ion have the necessary energy to disrupt the atomic bonding of the crystal and moves freely. We claim that this argument is reasonable for energies up to 60 eV (recoil energies).

The basic integral equation deduced by Lindhard [14] takes into account, in this case for atomic motion $\bar{\nu}$; that the atomic motion energy given by the ion with energy $E$ is equal to the atomic motion energy given by the ions after scattering, where one of them takes in to account ionization and the other target ion have to spend some energy in disrupting the binding,

$$\underbrace{\int d\sigma_{n,e}}_{\text{total cross section}} \left[ \underbrace{\bar{\nu}\left(E - T_n - \sum_i T_{ei}\right)}_{\text{All scatter-ions ions}} + \underbrace{\bar{\nu}(T_n - U)}_{\text{All Target ions}} + \underbrace{\bar{\nu}(E)}_{\text{Total initial energy}} + \underbrace{\sum_i \bar{\nu}_e(T_{ei} - U_{ei})}_{\text{Electrons contribution}} \right] = 0. \tag{2}$$

[1]Dont confuse with Lindhard parametrization model for ionization efficiency.

Where $T_n$ is the nuclear recoil energy given by the incident ion to the ion at rest, $\sum_i T_{ei}$ is the energy transfer to electron during the ion colision, $U$ is the atomic binding energy, $\sigma_{n,e}$ is the inelastic nuclear ion-ion corss section and is a fucntion of $T_n$ and $\sum_i T_{ei}$ and $\bar{\nu}_e$ is the atomic motion due to electrons (negligible). As we can see at low energies the part of atomic motion given by electron can be neglected. Lindhard gives an asymptotic approximate solution to Eq.(2) that is just valid at high energies.[2] Lindhard approximate solution works fine in many cases, e.g. Si, Ge, etc, for energies above 10 keV, but already for Si this parametrization fails below 4 keV [7].

## 3 Simplify Integro Differential Equation

In order to handle the integral equation Eq.(2), we make the following basic assumptions (most of them made by Lindhard); I we consider that nuclear recoil energy and the energy given to electrons are small compared to the initial kinetic energy of the ion $E$, II effects of electronic and atomic collisions can be treated separately. *In contrast to Lindhard we are not going to neglect the effects of binding energy.*

Furthermore, in order to get a solvable integro-differential equation, we need to relax approximation I up to second order. Also we consider nuclear stopping power using the universal nuclear cross section, that can have some variations depending on the inter atomic potentials used. With this and other details we can deduce a second order integro differential equation, that can be solve by using the shooting method [6],

$$-\frac{1}{2}k\varepsilon^{3/2}\bar{\nu}''(\varepsilon) + k\varepsilon^{1/2}\bar{\nu}'(\varepsilon) = \int_{\varepsilon u}^{\varepsilon^2} dt \frac{f(t^{1/2})}{2t^{3/2}}[\bar{\nu}(\varepsilon - t/\varepsilon) + \bar{\nu}(t/\varepsilon - u) - \bar{\nu}(\varepsilon)], \qquad (3)$$

where, $t = \varepsilon^2 \sin^2(\theta/2)$ (center of mass frame), $\sigma_n$ is the nuclear scattering cross section and, $S_e = k\varepsilon^{1/2}$ is electronic stopping power of the medium, if the electronic stopping is zero at all energies, then the quenching factor also might be zero. This equation predicts an energy threshold at $2u$. By firt considering a constant binding energy model and the electronic stopping power given by Lindhard, we succes in describe Si and Ge measuremnts at low energies [6].

## 4 Low and High energy Effects for Electronic Stopping Power

### 4.1 High energies effects

One of the limitations of the aforementioned approach in Si, relies in having a cut off too high compared to the expected threshold given by the energy to create a Frenkel-pair ($\approx 30$ eV) [8]. To affront this limitations, in this study we considered a varying binding energy model and Coulomb repulsion effects for low energies and electron stripping at higher energies. Hence, Lindhard electronic stopping is not valid at low energies ($\approx 5 <$keV).

Also we are going to considered electronic straggling effects in to the cascade process, this is done by expanding at second order in $\sum_i T_{ei}$ the term $\bar{\nu}(E - T_n - \sum_i T_{ei})$ in Eq.(2). The main effect of including electronic straggling is to low the quenching factor near threshold and slightly increase it at high energies. These effects can be added directly into Eq.(3).

---

[2]$f_n = kg(\varepsilon)/(1 + kg(\varepsilon)), \quad g(\varepsilon) = \varepsilon + 3\varepsilon^{0.15} + 6\varepsilon^{0.7}$.

For high energy the electronic stopping has to be corrected by velocity effects that reduce the effective number of electron in the ions, this can be modeled by using the Bohr stripping criteria [15]. The oscillatory effects for electronic stopping as function of the number of electrons of the incoming ion are very well known. Since due to Bohr stripping the effective number of electrons reduce, implies that for energies much lower than the Bragg peak (<20 Mev in Si) this effect may be active. This oscillations are caused by an appearance of a phase shift,Firedel sum rule [16], to maintain neutrality of electron Fermi gas, it can be tested that this effect produce a better agreement with the model and the available data for electronic stopping in silicon.

## 4.2  Low energies effects

At low energies, it is very well known that the electronic stopping tents to damp in a non proportional velocity dependence [17], due to at very low relative velocities (compared to $v_0$) colliding nuclei will not penetrate the electron clouds of each other strongly, this defines the Coulomb repulsion effects for electronic stopping power.

The general formula for electronic stopping according kinetic theory [18]

$$S_e = (\Xi)Nmv \int_R^\infty v_F \sigma_{tr}(v_F)N_e dV \, , \tag{4}$$

where $R$ is the distance of closest approach, that is computed by solving $E = \frac{1}{2}V(R)$ for a given inter-atomic potential, $\sigma_{tr}$ is the transport cross section, $N_e$ the electron density, $v_F$ the Fermi velocity of the electron gas and $\Xi$ a geometrical factor previously mentioned by Tilinin [9]. This factor is only relevant for high atomic number elements, like Ge, where for Si the effect can be neglected and approximate $\Xi \approx 1$. Here we consider three different models for the analysis, Tilinin [9] (Transport cross section and kinetic theory), Kishinevsky [19] (based on Firsov model with inter-atomic interaction included ) and Arista [20] (based on the dielectric function formalism, based in Lindhard ideas). Furthermore we consider four inter-atomic potentials;Thomas-Fermi, Moliere, AVG and Ziegler [21,22]. Both the scale factor $\xi_e$ and the inter-atomic potential used affects the electronic stopping and the binding energy, specially at low energies.

For the binding energy model we include the Frenkel energy, i.e. the energy to create a free ion in the crystal lattice, mainly important at low energies, and we include inner excitation of the electron clouds, that are mainly important at high energies. In general high binding energies, tent to reduce the quenching factor.

Usually Density Functional Theory is used to model electronic stopping and binding energies. The kinetic energy for electrons usually is taken to be the average free electron energy of a Fermi gas $(3/5)E_F$ where this assumption may work at high energies. But for low energies interaction, Tilinin [9] makes the observation that only electrons near the Fermi energy can be excited, due to Pauli exclusion principle, since inner electron occupy the energy levels. So if we take in to account this observation, we can change in the model that the average electron kinetic energy to be just $E_F$. By doing this it can be shown that the atomic scale change by a factor of 5/3. This implies for the electronic stopping the appearance of an scaling factor of $\xi = 2.15$. Before this Lindhard observed the need to ad a corrective factor to electronic stopping in the range among one or two. This argument may give a reasonable explanation about the physical origin of this scale factor in the context of density functional theory. In this study we are going to consider this atomic scaling factor $\xi$, as a free parameter among the range 1 to 2.15, furthermore this also affects the binding energy model and nuclear recoils interactions.

## 5 Ionization Efficiency for Noble Gases

For noble gases we can apply our model for the quenching factor for LXe and LAr, using recent measurements of the ionization yield $Q_y^{\text{ER}}$ and the scintillation efficiency $L_y$ [23]. Reconstruction is done by exploiting the full anticorrelation between the S1 (scintillation photons $n_\gamma$) and S2 (ionized electrons $n_e$), where it is usually to assume that each excited or ionized atom leads to one scintillation photon or electron, hence for the total quanta defined by the number of ions $N_i$ and excitons $N_{ex}$ produced, $N_i + N_{ex} = n_\gamma + n_e$ independent of recombination.

The fraction of ionizations due to recombination is predicted by the Thomas-Imel box model [24].This model has been shown to work well for spatially small tracks. The charge yield is proportional to Eq.(1) by the electron recoil energy $E_{er} = f_n E_R$ and the energy $W_i$ to create an electron-hole pair in the liquid, $N_i = f_n(\frac{E_R}{W_i})$

$$Q_y^{\text{ER}} = \frac{N_{\text{i}}}{E_{\text{er}}} = \frac{(1-r)N_i}{E_{\text{er}}}, \quad 1-r = \frac{1}{\gamma N_i} \ln(1 + \gamma N_i), \tag{5}$$

where $\gamma$ is a free parameter of the model typically of the order of $10^{-2}$. In analogy for the light yield,

$$L_y = \frac{N_i(r + N_{ex}/N_i)}{E_{er}}, \tag{6}$$

where $N_{ex}/N_i$ will be considered as a constant, the typical values are of the order of $\approx 1/2$. We compute the charge and light yields for LAr and LXe, using the constant binding energy model and $S_e = k\varepsilon^{1/2}$, as a first attempt to explain recent low energies measurements.

## 6 Results and Applications

We show in Fig.(1), the preliminary result for Si, compared with the constant binding energy model and Lindhard's. Now we can describe five orders of magnitude of data. We fit the inter-atomic scale parameter to electronic stopping power ($S_e$) data for Si-Si ions [25–27], that scales the electronic and nuclear stopping and has important effects at low energies. In this study we obtained $\Xi = 1.46$ to be a best fit for $S_e$ data. Even though the model is independent of Lindhard's, we see a very good match with our model at high energies.

Excess signals produced by flat background can be expected to appear at low energies due to new effects, binding energy, considered by Eq.(3) than Lindhard's model fail to reproduce. For flat nuclear recoil spectrum or background signals, e.g. thermal neutrons, we can explain an EXCESS signal considering that typically Monte Carlo (MC) simulations used Lindhard's theory to reconstruct from recoil to visible energy in experiments. If in substitution we use our model for quenching factor, we reconstruct a signal that have an EXCESS at low energy compared from this MC simulations. The physical origin of this excess relies in the inter-atomic interactions due to binding energy (energy stored in a defect of the crystal), electronic and nuclear stopping.

Defect production in crystals is a recent are of study [28], that has potential to be applied for low energy detection. With our model and the Kinchin and Pease theory [29] , is very straight forward to compute the number of Frenkel pairs in a crystal in Si. We note that the expected number of events for our model is great than the prediction by using Lindhard's model.

For germanium we can use Eq.(4) to compute the electronic stopping power and introduce it to Eq.(3) (with straggling) for $f_n$ computation. But as mentioned before Tilinin approach fails for large atomic numbers like Ge. In this case it is necessary to introduced and model

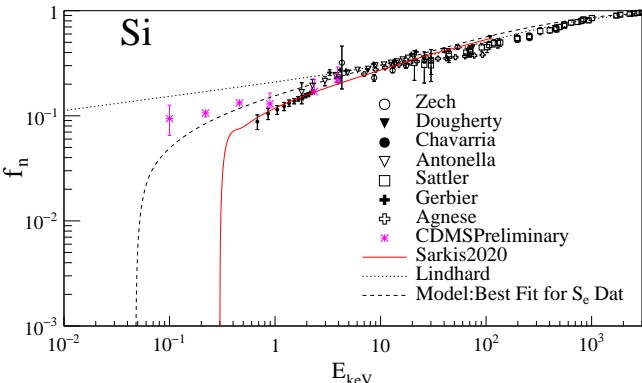

Figure 1: Silicon quenching factor measurements points taking from [6,32–34]. The points are compared with, (line dotted) the solution of Eq.(3) with new effects added at low and high energies, (dotted) Lindhards, and (red)the solution of Eq.(3) with constant binding energy and Lindhard electronic stopping power.

the geometrical factor $\Xi$ that may appear in Eq.(4). This can be modeled by considering the electron ionization cross section, the rate of electrons emitted by collision and trajectory different from a straight line of scattering. The model is still in progress and under revision, here we show the preliminary results in Fig.(3).

Finally we show in fig the results for Charge and light yield in LXe and LAr, see figures from [30]. We fit the parameters for Thomas Imel box model, giving $\gamma_{Xe} = 0.0127$, $\gamma_{Ar} = 0.025$, $(N_{ex}/N_i)_{Xe} = 0.47$ and $(N_{ex}/N_i)_{Ar} = 0.687$. The results are compare to measurements taken from [23] and NEST semi-empirical model.

## 7 Conclusions

We present a general model based on integral equations for ionization in pure crystals and noble liquids. The model predicts the turnover of $f_n$ at low energies, already observed in Xe for $E_R < 1$ keV. We incorporate corrections due to electronic straggling and atomic scaling in the Int. Diff. Eq. For silicon Coulomb effects allow us to fit the data up to 3 MeV and have a threshold near Frenkel-pair creation energy. For germanium our model shows potential to explain recent measurements [31]. We show charge and light yields for LXe and LAr consistent with actual data. Much work can be done from here, e.g directional quenching factor, straggling for $\bar{\nu}$, etc.

## Acknowledgements

We specially thank to Daniel Baxter from Fermilab for presenting this work in the EXCESS-2022 conference at Viena in behalf of the authors.

**Funding information** This research was supported in part by DGAPA-UNAM grant number PAPIIT-IT100420, PAPITT-IN106322 and Consejo Nacional de Ciencia y Tecnología (CONA-CYT) through gran CB2014/240666.

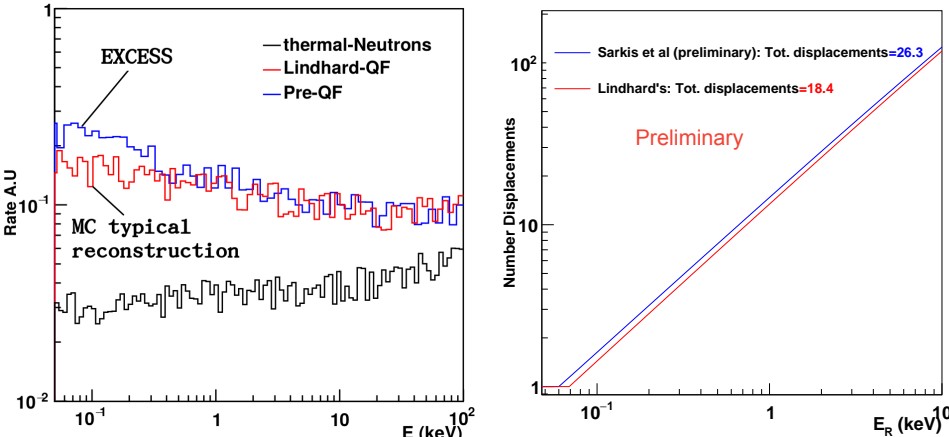

Figure 2: (left) Excess signal predicted from this work compared to Lindhard's expectations. (right) Number of defects from this work (blue) compared to Lindhard model (red).

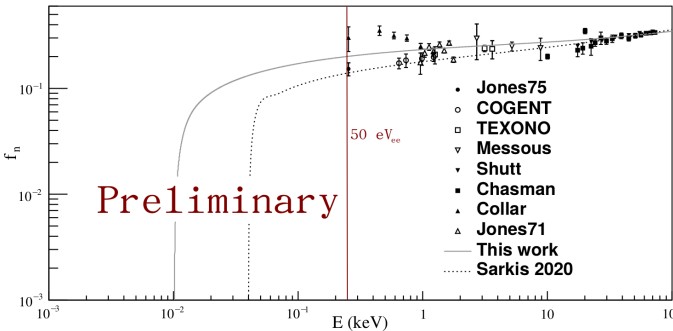

Figure 3: Germanium preliminary quenching factor computation based on the solution of Eq. (3), where the geometrical factor mentioned by Tilinin is considered.

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
