# Peer review of "Ionization efficiency at sub-keV energies for crystals and noble liquids"

_SciPost Physics Proceedings, doi:SciPost Phys. Proc. 12, 008 (2023)_

## Round 1 · Referee Report · Anonymous · 2022-11-8

Strengths

Detailed study with possible new effects about the ionization and light yields produced by nuclear recoils at low energies in crystals and noble liquids. Authors consider three different models for the electronic stopping power taking to account more effects. The work is done in the frame of EXCESS workshop which was triggered by recent unexplained rise of events at low energies.

Weaknesses

typos and inconsistency in writing, requires careful formatting

Report

The manuscript meets criteria. I recommend to publish it after some corrections.

Requested changes

- in introduction you say "pure crystals" - be more specific what do you mean
-" by the an ion" the or an?
- takes in to account -> into
- formula (2) please explain variables for clarity
- in introduction you state that Lindhard equation describes data for Si below 4keV accurately, later you state that it fails. Can you make it more clear?
- "most of then"? or them?
- CM frame?
- "Low and high energy effects for Se"? Se - write with words
- the main facts ignored?
- we are going to consider?
- give numbers to all formulas

  • validity: -
  • significance: high
  • originality: high
  • clarity: good
  • formatting: good
  • grammar: good

Author:  Youssef Sarkis  on 2022-11-16  [id 3035]

(in reply to Report 1 on 2022-11-08)
Category:
answer to question
correction

  • in introduction you say "pure crystals" - be more specific what do you mean: We have add instead Si and Ge ionization detectors. -" by the an ion" the or an?: was change "by the ion" -takes in to account -> into: changed -formula (2) please explain variables for clarity: we have put some description to the relevant quantities that appear in Eq. 2. -in introduction you state that Lindhard equation describes data for Si below 4keV accurately, later you state that it fails. Can you make it more clear?: Here not confuse Lindhard integral equation with Lindhard parametrization model, are different. For the sake of clearness, we have added a footnote and giving more emphasis that one is Lindhard parametrization and the other is the numerical solution of Lindhard integral equation with constant binding energy.
    "most of then"? or them?: them, changed.
  • CM frame?: changed by center of mass.
  • "Low and high energy effects for Se"? Se - write with words: done
  • the main facts ignored?: changed to "To affront this limitations"
  • we are going to consider?: changed to we considered.
  • give numbers to all formulas: done

---

## Round 2 · List of Changes

in introduction you say "pure crystals" - be more specific what do you mean: We have add instead Si and Ge ionization detectors. -" by the an ion" the or an?: was change "by the ion" -takes in to account -> into: changed -formula (2) please explain variables for clarity: we have put some description to the relevant quantities that appear in Eq. 2. -in introduction you state that Lindhard equation describes data for Si below 4keV accurately, later you state that it fails. Can you make it more clear?: Here not confuse Lindhard integral equation with Lindhard parametrization model, are different. For the sake of clearness, we have added a footnote and giving more emphasis that one is Lindhard parametrization and the other is the numerical solution of Lindhard integral equation with constant binding energy.
"most of then"? or them?: them, changed.
CM frame?: changed by center of mass.
"Low and high energy effects for Se"? Se - write with words: done
the main facts ignored?: changed to "To affront this limitations"
we are going to consider?: changed to we considered.
give numbers to all formulas: done
"most of then"? or them?: them, changed.
CM frame?: changed by center of mass.
"Low and high energy effects for Se"? Se - write with words: done
the main facts ignored?: changed to "To affront this limitations"
we are going to consider?: changed to we considered.
give numbers to all formulas: done

---

## Editorial Decision

published